# Platelet Concentration and Platelet/Lymphocyte Ratio as Prognostic Indicators in Luminal Breast Cancer

**Angela Della'Santa Rubio O. Rönnau [1], Maiquidieli Dal Berto [2], Claudia Giuliano Bica [2], Rafael Vargas Alves [2,3]**
**and Liane Nanci Rotta [1,*]**

[1] Graduation Program in Health Sciences, Federal University of Health Sciences of Porto Alegre,
  Porto Alegre 90050-170, RS, Brazil
[2] Laboratory of Pathology Research, Graduation Program in Pathology, Federal University of Health Sciences
  of Porto Alegre, Porto Alegre 90050-170, RS, Brazil
[3] Santa Casa de Misericórdia de Porto Alegre, Porto Alegre 90050-170, RS, Brazil
[*] Correspondence: lnrotta@ufcspa.edu.br

**Abstract:** Ratios between the blood cells are indirect measures of the imbalance in the pro-inflammatory status observed in carcinogenesis and have been proposed as accessible and feasible biomarkers to predict cancer prognosis. We aim to evaluate the prognostic significance of neutrophil/lymphocyte (NLR), monocyte/lymphocyte (MLR), and platelet/lymphocyte (PLR) ratios in Brazilian patients with luminal breast cancer (LBC) treated with tamoxifen. A retrospective cohort of 72 operable LBC patients. Preoperative leukocyte and platelet absolute values permitted to calculate NLR, MLR, and PLR. Area under curve (ROC) determined the cutoff value associated with relapse and death. Univariate and multivariate analyses were used to assess the relationship of the platelet and PLR to disease-free survival (DFS) and overall survival (OS). Lower DFS was associated with $>297 \times 10^3/mm^3$ (54 vs. 60.9 months in <297, $p = 0.04$). Platelet $> 279 \times 10^3/mm^3$ are related to higher OS ($p = 0.03$). Univariate analysis revealed that platelet concentration was associated with DFS ($p = 0.04$) and OS ($p = 0.04$), but not as an independent factor (HR = 1.31, 95%CI: 0.42–4.07, $p = 0.65$) and OS (HR = 1.64, 95%CI: 0.28–9.52, $p = 0.58$). Both univariate ($p = 0.01$) and multivariate analysis revealed that PLR < 191.5 was a significant independent predictor of higher OS/better prognosis (HR = 16.16, 95%CI: 2.83–109.25, $p = 0.00$). Pretreatment platelet indices (absolute count and PLR) are prognosis predictors in LBC patients. Platelet $> 279 \times 10^3/mm^3$ and PRL < 191.5 was associated with a higher OS, with the PRL being an independent predictor of higher OS.

**Keywords:** luminal breast cancer; prognosis; platelet; platelet/lymphocyte ratio

## 1. Introduction

Breast cancer (BC) is a complex disease, with high clinical, morphological, and biological heterogeneity. Breast tumors with similar histology and clinic may have different prognoses and therapeutic responses. In women, BC is the most common malignant tumor, being one of the leading causes of cancer death. Its prognosis and treatment are defined by location, age of presentation, and staging, as well as risk factors that consider hormonal, biological, molecular, and genetic criteria [1–3].

The evolution of cancer reflects the complex cellular and molecular interactions between the tumor and the host's immune system, promoting mutations or sustaining DNA damage, allowing its proliferation in environments rich in inflammatory cells and growth factors [4]. Thus, the chronic inflammatory response is associated with tumor initiation and promotion [5]. The greater the local inflammatory reaction induced by the tumor, the greater will be the tumor's aggression and its ability to spread over long distances, reaching blood vessels and lymph nodes, generating metastases [6].

Several factors affect the BC prognosis, including clinic pathological features (patient age, lymph node status, tumor size) and molecular biology parameters (hormonal receptors,

human epidermal growth factor receptor 2, HER2, and molecular subtype) [4,7]. Leukocytes associated with the tumor can act in defense and contribute to tumor progression, due to the inhibitory and stimulating functions that they can exert on neoplastic cells. Evidence shows that the evaluation of the expression of neutrophils, lymphocytes, monocytes, and platelets in peripheral blood adds prognostic information to the traditionally used criteria, including the prediction of metastasis [8,9].

Neutrophil-to-lymphocyte (NLR), monocyte-to-lymphocyte (MLR), and platelet-to-lymphocyte (PLR) ratios have been indicated as predictors of prognosis in some types of cancers and are currently recommended as auxiliary tools in the definition of indication for adjuvant treatment [10–13]. We aim to evaluate these blood cellular ratios in Brazilian patients with luminal breast cancer (LBC) treated with tamoxifen, as general measures of the inflammatory process and immune status and correlated with the prognosis (disease-free survival, DFS, and overall survival, OS).

## 2. Materials and Methods

The retrospective cohort study analyzed 72 patients diagnosed with LBC A and B, with convenience sampling. The inclusion criteria considered female patients diagnosed at Santa Casa de Misericordia Hospital Complex in Porto Alegre/Brazil (2007–2014), aged $\geq$ 18 years, with the requirement of diagnosing BC with positive hormone receptors. Only patients who used tamoxifen and who had blood count results prior to the treatment were included in the hospital's computerized system. All patients were followed up to 6 years in the study. Male patients or those under 18 years old were excluded, as well as patients diagnosed with negative hormone receptors, who did not use tamoxifen, with ongoing infection or autoimmune disease, or using steroid drugs and immunosuppressants.

The data were obtained from clinical records and include demographic variables, diagnosis date, adjuvant therapy, dates of initiation of therapy/surgery/beginning of radiotherapy, tumor size (stage T), the status of lymph nodes (stage N), and metastases. For the performance of cellular blood analysis, a preoperative blood sample closest to the date of surgery (minimum: 1 day and maximum: 90 days) was used and the absolute values of neutrophils, lymphocytes, platelets, and monocytes were collected (obtained by automatized hematological technologies comparable during the full period of the study), which were used to establish the ratio between cells and the systemic inflammation index (SII). SII data were calculated using the formula P (platelets) $\times$ N (neutrophils)/L (lymphocytes); NLR was defined as N (neutrophils)/L (lymphocytes); MLR as M (monocytes)/L (lymphocytes), and PLR as P (platelets)/L (lymphocytes). Disease-free survival (DFS) was calculated from the date of diagnosis of the pathology until the date of local or distant recurrence, death, or new cancer. Overall survival (OS) was calculated from the date of diagnosis of the pathology until death (from any cause) or the date of the last follow-up (June 2019).

The Ethics Committee approved this study (CAAE 22396713.1.0000.5335), and all experimental procedures were performed according to the Declaration of Helsinki. All participants were informed about the study and signed the informed consent.

Data analysis was performed using the Statistical Package for Social Sciences (SPSS) version 23.0 for Windows®. A $p < 0.05$ was considered statistically significant. To characterize the sample, categorical variables were demonstrated according to their frequency and percentage and quantitative variables were described with mean and standard deviation. The chi-square test or Fisher's test was used to assess the differences in the distribution of the categorical variables analyzed. For the univariate and multivariate analyses, the Cox proportional hazards regression model with a 95% confidence interval was used. The ROC curve was plotted to assess the sensitivity and specificity of the ratios, as well as the SII and absolute cell counts, in comparison with the outcomes (relapse and death).

## 3. Results

### 3.1. Characteristics of the Patients

In total, 400 BC women were evaluated, of which 160 were classified as LBC, and 72 patients with the diagnosis of Luminal A or B breast cancer who received tamoxifen for 5 years were included in the study. The disease incidence was 42% at 40 to 60 years, 44% < 40 years, and 14% > 60 years. The mean age for patients with relapse was $46.1 \pm 12.6$ years (Tables 1 and 2). Ductal carcinoma (87.5%), tumor grade II (48.6%), and stage II (40.3%) were predominant and Luminal A and Luminal B were estimated in 47 patients (65.3%) and 25 patients (34.7%), respectively (Table 1).

**Table 1.** Characterization of the patients according to relapse and death.

| | | Relapse | | Death | | |
|---|---|---|---|---|---|---|
| **Characteristic** | **N (%)** | **Without 46 (63.9)** | **With 26 (36.1)** | **Without 59 (81.9)** | **With 13 (18.1)** | **Total 72 (100)** |
| Age at diagnosis (72) | | $47.0 \pm 12.2$ | $46.1 \pm 12.6$ | $47.3 \pm 12.8$ | $44.2 \pm 9.88$ | |
| ≤40 year | 32 (44) | 19 (41.3) | 13 (50.0) | 25 (42.4) | 7 (53.8) | 32 (44.4) |
| 40 to 60 years | 30 (42) | 20 (43.5) | 10 (38.5) | 25 (43.5) | 5 (38.5) | 30 (41.7) |
| ≥60 years | 10 (14) | 7 (15.2) | 3 (11.5) | 9 (15.3) | 1 (7.7) | 10 (13.9) |
| Tumoral grade (72) | | | | | | |
| I | 12 (16.7) | 11 (23.9) | 1 (3.8) * | 12 (20.3) | 0 (0.0) | 12 (16.7) |
| II | 35 (48.6) | 24 (52.2) | 11 (42.3) * | 29 (49.2) | 6 (46.2) | 35 (48.6) |
| III | 25 (34.7) | 11 (23.9) | 14 (53.8) * | 18 (30.5) | 7 (53.8) | 25 (34.7) |
| Clinical staging (71) | | | | | | |
| T1 | 27 (37.5) | 23 (50.0) | 4 (15.4) ** | 26 (44.8) | 1 (7.7) * | 27 (38.0) |
| T2 | 29 (40.3) | 19 (41.3) | 10 (38.5) ** | 24 (41.4) | 5 (38.5) * | 29 (40.8) |
| T3/T4 | 15 (20.8) | 4 (8.7) | 11 (42.3) ** | 8 (13.8) | 7 (53.8) * | 15 (21.2) |
| N0 | 36 (50.0) | 32 (69.6) | 4 (15.4) ** | 33 (56.9) | 3 (23.1) * | 36 (50.7) |
| N1 | 20 (27.8) | 12 (26.1) | 8 (30.8) ** | 16 (27.6) | 4 (30.8) * | 20 (28.2) |
| N2/N3 | 15 (20.8) | 2 (4,4) | 13 (50.0) ** | 9 (15.5) | 6 (46.1) * | 15 (21.1) |
| Treatment | | | | | | |
| Radiotherapy | 62/69 (86.1) | 39 (62.9) | 23 (37.1) | 49 (87.5) | 13 (100) | 62/69 (89.8) |
| Chemotherapy | 16/61 (22.2) | 27 (44.3) | 18 (29.5) ** | 36 (69.2) | 9 (100) | 45/61 (73.8) |
| Histology (72) | | | | | | |
| Ductal | 63 (87.5) | 41 (65.1) | 22 (34.9) | 52 (82.5) | 11 (17.4) | 63 (87.5) |
| Lobular | 9 (12.5) | 5 (55.5) | 4 (44.4) | 7 (77.8) | 2 (22.2) | 9 (12.5) |
| Immunohistochemistry (72) | | | | | | |
| Ki 67 < 14 | 45/72 (62.5) | 33 (73.3) | 12 (26.7) | 39 (66.1) | 6 (46.2) | 45 (62.5) |
| Ki 67 > 14 | 27/72 (37.5) | 14 (51.8) | 13 (48.2) | 20 (33.9) | 7 (53.8) | 27 (37.5) |

Chi-square (Pearson) * $p < 0.05$; ** $p < 0.01$.

**Table 2.** Characteristics of patients according to platelet concentration—relapse outcome.

| **Variable** | **N** | **Platelet ($\times 10^3$/mm$^3$)** | | *p* |
|---|---|---|---|---|
| | | **<297** | **>297** | |
| Age at diagnosis | 72 | 43 (100) | 29 (100) | 0.35 |
| ≤40 years | 32 | 21 (48.8) | 11 (37.9) | |
| 40 to 60 years | 30 | 18 (41.9) | 12 (41.1) | |
| ≥60 years | 10 | 4 (9.3) | 6 (20.7) | |
| Median ± std deviation | | $45.2 \pm 11.7$ | $49.1 \pm 13.0$ | 0.19 |
| Tumoral grade | 72 | 43 (100) | 29 (100) | 0.08 |
| I | 12 | 10 (23.3) | 2 (6.9) | |
| II | 35 | 22 (51.2) | 13 (44.8) | |

**Table 2.** *Cont.*

| Variable | N | Platelet ($\times 10^3$/mm$^3$) | | p |
|---|---|---|---|---|
| | | **<297** | **>297** | |
| III | 25 | 11 (25.6) | 14 (48.3) | |
| Clinical staging | 71 | 42 (100) | 29 (100) | |
| T1 | 27 | 21 (50.0) | 6 (20.7) | 0.01 |
| T2 | 29 | 16 (38.1) | 13 (44.8) | |
| T3 | 7 | 4 (9.5) | 3 (10.3) | |
| T4 | 8 | 1 (2.4) | 7 (24.1) | |
| N0 | 71 | 24 (57.1) | 12 (41.4) | 0.01 |
| N1 | 20 | 15 (35.7) | 5 (17.2) | |
| N2 | 8 | 3 (7.1) | 5 (17.2) | |
| N3 | 7 | 0 (0.0) | 7 (24.1) | |
| Relapse | 72 | 43 (100) | 29 (100) | 0.04 |
| No | 46 | 32 (69.6) | 14 (30.4) | |
| Yes | 26 | 11 (42.3) | 15 (57.7) | |
| Death | 72 | 43 (100) | 29 (100) | 0.04 |
| No | 59 | 39 (66.1) | 20 (33.9) | |
| Yes | 13 | 4 (30.8) | 9 (69.2) | |
| Radiotherapy | 69 | 40 (100) | 29 (100) | 0.69 |
| No | 7 | 5 (12.5) | 2 (6.9) | |
| Yes | 62 | 35 (87.5) | 27 (93.1) | |
| Chemotherapy | 61 | 39 (100) | 22 (100) | 0.44 |
| No | 16 | 12 (30.8) | 4 (18.2) | |
| Yes | 45 | 27 (69.2) | 18 (81.8) | |
| Histology | 72 | 43 (100) | 29 (100) | 0.73 |
| Ductal | 63 | 37 (86.0) | 26 (89.7) | |
| Lobular | 9 | 6 (14.0) | 3 (10.3) | |
| Leukocyte | 72 | 6124 ± 2057 | 7355 ± 1986 | 0.01 |
| Neutrophil | 72 | 3647 ± 1419 | 4605 ± 1635 | 0.01 |
| Lymphocyte | 72 | 1788 ± 743 | 1969 ± 650 | 0.29 |
| Monocyte | 72 | 483 ± 176 | 576 ± 154 | 0.02 |
| 6 years DFS (95% CI) | 72 | 60.9 (54.7–67.2) | 54.0 (46.3–61.6) | 0.04 |
| 6 years OS (95% CI) | 72 | 119.1 (113.6–124.6) | 127.4 (110.1 ± 144.7) | 0.08 |

The results are expressed as n (%), except for absolute count of leukocytes, neutrophils, lymphocytes, monocytes, and platelets ($\times 10^3$/mm$^3$), that are expressed as median ± standard deviation. DFS: disease-free survival; OS: overall survival, expressed in months.

The characterization of patient samples in relation to disease relapse and death shows a difference in histological grade ($p < 0.05$) and staging (T and N) ($p < 0.01$). During the study, 13 (18.1%) patients died, who were in the advanced stages ($p < 0.05$) (Table 1).

### 3.2. Establishment of Cutoff—ROC Curves

The ROC curve was performed to determine the cutoff value of the studied parameters. For the relapse outcome, the platelet cutoff value of $297 \times 10^3$/mm$^3$ was defined

(AUC = 0.650, 58% sensitivity, and 70% specificity) (95%CI: 0.487–0.0752, $p$ = 0.07). For the death outcome, the value for PLR was 191.5 (AUC = 0.700, 54% sensitivity, and 78% specificity) (95% CI: 0.561–0.839, $p$ = 0.02), of platelets was $279 \times 10^3/mm^3$ (AUC = 0.662, 85% sensitivity, and 56% specificity) (95%CI: 0.497–0.828, $p$ = 0.07), and the SII was 431.3 (AUC = 0.623, 100% sensitivity, and 36% specificity) (95%CI: 0.478–0.769, $p$ = 0.17) (data not shown). All other studied parameters showed AUC < 0.65, not allowing the establishment of a cutoff value.

Table 2 shows the baseline characteristics of the patients according to the platelet count considering the outcome of relapse. Platelet values ($\times 10^3/mm^3$) < 297 were observed in the lower clinical stages of T1 and T2 (50 and 38.1%) and N0 and N1 (57.1 and 35.7%). Approximately 70% of patients who did not relapse and did not die had platelet counts <297, while 57.7% and 69.2% of those who relapsed and died, respectively, showed platelet counts >297. A higher absolute cell count of leukocytes ($p$ = 0.01), neutrophils ($p$ = 0.01), and monocytes ($p$ = 0.02) were associated with platelets >$297 \times 10^3/mm^3$. Lower DFS was associated with a higher platelet number (54 months at >297 vs. 60.9 months at <297).

Table 3 shows the characteristics of the patients according to the RPL, platelets ($\times 10^3/mm^3$), and SII considering the death outcome. Of the 52 patients with PLR < 191.5, 6 (11.5%) died, while the 20 patients with PLR > 191.5, 7 (35.0%) died ($p$ = 0.04). Except for the absolute monocyte count, all other absolute cell counts showed higher values when RPL < 191.5 and SII > 431.3. When the platelets were <279, 91.2% of the patients had T1/T2 and N0/N1 staging, while 67.5% of the patients with platelets >279 had the same staging; 5.7% and 29.7% of the patients died when the platelets were <279 and >279, respectively ($p$ = 0.02). No death was observed in patients who presented SII < 431.3, while 25.5% of patients died when SII > 431.3.

**Table 3.** Characteristics of patients, according to PLR, platelet count, and systemic index of inflammation (SII)—death outcome.

| Variable | N | PLR | | $p$ | Platelet ($\times 10^3/mm^3$) | | $p$ | SII | | $p$ |
|---|---|---|---|---|---|---|---|---|---|---|
| | | <191.5 | >191.5 | | <279 | >279 | | <431.3 | >431.3 | |
| Age at diagnosis | 72 | 46 (100) | 26 (100) | 0.36 | 35 (100) | 37 (100) | 0.74 | 21 (100) | 51 (100) | 0.78 |
| ≤40 years | | 25 (48.1) | 7 (35.0) | | 17 (48.6) | 15 (40.5) | | 10 (47.6) | 22 (43.1) | |
| 40 to 60 years | | 19 (36.5) | 11 (55.0) | | 14 (40.0) | 16 (43.2) | | 9 (42.9) | 21 (41.2) | |
| ≥60 years | | 8 (15.4) | 2 (10.0) | | 4 (11.4) | 6 (16.2) | | 2 (9.5) | 8 (15.7) | |
| Tumoral grade | 72 | 52 (100) | 20 (100) | 0.61 | 35 (100) | 37 (100) | 0.12 | 21 (100) | 51 (100) | 0.82 |
| I | | 10 (19.2) | 2 (10.0) | | 7 (20.0) | 5 (13.5) | | 4 (19.0) | 8 (15.7) | |
| II | | 24 (46.2) | 11 (55.0) | | 20 (57.1) | 15 (40.5) | | 9 (42.9) | 26 (51.0) | |
| III | | 18 (34.6) | 7 (35.0) | | 8 (22.9) | 17 (45.9) | | 8 (38.1) | 17 (33.3) | |
| Clinical staging | 71 | 51 (100) | 20 (100) | | 34 (100) | 37 (100) | | 20 (100) | 51 (100) | |
| T1 | | 20 (39.2) | 7 (35.0) | 0.09 | 17 (50.0) | 10 (27.0) | 0.02 | 7 (35.0) | 20 (39.2) | 0.38 |
| T2 | | 24 (47.1) | 5 (25.0) | | 14 (41.2) | 15 (40.5) | | 11 (55.0) | 18 (35.3) | |
| T3 | | 3 (5.9) | 4 (20.0) | | 3 (8.8) | 4 (10.8) | | 1 (5.0) | 6 (11.8) | |
| T4 | | 4 (7.8) | 4 (20.0) | | 0 (0.0) | 8 (21.6) | | 1 (5.0) | 7 (13.7) | |
| N0 | 71 | 24 (47.1) | 12 (60.0) | 0.54 | 24 (61.8) | 15 (40.5) | 0.04 | 11 (55.0) | 25 (49.0) | 0.34 |
| N1 | | 14 (27.5) | 6 (30.0) | | 10 (29.4) | 10 (27.0) | | 7 (35) | 13 (25.5) | |
| N2 | | 7 (13.7) | 1 (5.0) | | 3 (8.8) | 5 (13.5) | | 2 (10.0) | 6 (11.8) | |
| N3 | | 6 (11.8) | 1 (5.0) | | 0 (0.0) | 7 (18.9) | | 0 (0.0) | 7 (13.7) | |
| Relapse | 72 | 52 (100) | 20 (100) | 0.88 | 35 (100) | 37 (100) | 0.12 | 21 (100) | 51 (100) | 0.26 |
| No | | 34 (65.4) | 12 (60.0) | | 26 (74.3) | 20 (54.1) | | 16 (76.2) | 30 (58.8) | |
| Yes | | 18 (34.6) | 8 (40.0) | | 9 (25.7) | 17 (45.9) | | 5 (23.8) | 21 (41.2) | |

**Table 3.** *Cont.*

| Variable | N | PLR | | p | Platelet (×10³/mm³) | | p | SII | | p |
|---|---|---|---|---|---|---|---|---|---|---|
| | | <191.5 | >191.5 | | <279 | >279 | | <431.3 | >431.3 | |
| Death | 72 | 52 (100) | 20 (100) | 0.04 | 35 (100) | 37 (100) | 0.02 | 21 (100) | 51 (100) | 0.01 |
| No | | 46 (88.5) | 13 (65.0) | | 33 (94.3) | 26 (70.3) | | 21 (100) | 38 (74.5) | |
| Yes | | 6 (11.5) | 7 (35.0) | | 2 (5.7) | 11 (29.7) | | 0 (0.0) | 13(25.5) | |
| Histology | 72 | 52 (100) | 20 (100) | 0.70 | 35 (100) | 37 (100) | 1.00 | 21 (100) | 51 (100) | 0.71 |
| Ductal | | 46 (88.5) | 17 (85.0) | | 31 (88.6) | 32 (86.5) | | 18 (85.7) | 45 (88.2) | |
| Lobular | | 6 (11.5) | 3 (15.0) | | 4 (11.4) | 5 (13.5) | | 3 (14.3) | 6 (11.8) | |
| Leukocyte | 72 | 7048 ± 2060 | 5508 ± 1829 | 0.00 | 6426 ± 2050 | 6804 ± 2166 | 0.45 | 5839 ± 2405 | 6942 ± 1900 | 0.04 |
| Neutrophil | 72 | 4210 ± 1541 | 3574 ± 1595 | 0.12 | 3941 ± 1314 | 4121 ± 1795 | 0.63 | 2973 ± 1408 | 4470 ± 1430 | 0.00 |
| Lymphocyte | 72 | 2095 ± 679 | 1252 ± 308 | 0.00 | 1779 ± 789 | 1937 ± 623 | 0.35 | 2137 ± 900 | 1746 ± 584 | 0.03 |
| Monocyte | 72 | 531 ± 175 | 496 ± 170 | 0.44 | 495 ± 185 | 545 ± 159 | 0.22 | 508 ± 210 | 526 ± 157 | 0.68 |
| Platelet (×10³/mm³) | 72 | 266 ± 63 | 321 ± 81 | 0.00 | 222 ± 32 | 337 ± 52 | 0.00 | 242 ± 49 | 297 ± 74 | 0.03 |

Age and absolute cellular count (leukocyte, neutrophil, lymphocyte, monocyte, and platelet) are expressed as median ± SD (Student *t* test); the other items were analyzed by chi-square (Fisher's Exact test). DFS: disease-free survival; OS: overall survival, expressed in months.

### 3.3. Prognostic Analysis

In the univariate analysis, the factors related to DFS in LBC include Grade 3 ($p = 0.01$), T3 ($p = 0.01$) and T4 ($p = 0.00$) staging, N1 ($p = 0.02$), N2 e N3 ($p = 0.00$), and platelets $> 297 \times 10^3/\text{mm}^3$ ($p = 0.04$). The multivariate analysis revealed that the platelet count did not remain as an independent predictor of DFS (HR = 1.31, $p = 0.65$). To assess OS based on the univariate analysis, the associated prognostic factors were the clinical staging T3 ($p = 0.02$), T4 ($p = 0.03$), and N3 ($p = 0.02$), PLR > 191.5 ($p = 0.01$), and platelets $> 279 \times 10^3/\text{mm}^3$ ($p = 0.04$). The multivariate analysis revealed that RPL remained as an independent predictor of OS (HR: 16.15, $p = 0.00$). We did not find differences in the 6-year OS between patients with platelets < or >297 (119.1 and 127.4 months, respectively) ($p = 0.08$) (Table 4).

Kaplan–Meier curves from DFS and OS in LBC, according to the platelet count ($\times 10^3/\text{mm}^3$), RPL, and SII, are shown in Figure 1. For all LBC patients, the DFS at 6 years was 58.13 ± 2.5 months (95%CI: 53.22–63.05) and the OS was 137.08 ± 5.02 months (95%CI: 127, 23–146.93). The platelet count showed an OS of 121.67 (95%CI: 117.30–126.04) and 127.21 (95% CI: 111.75–142.68) months for values <279 and $>279 \times 10^3/\text{mm}^3$, respectively ($p = 0.03$). The DFS, for platelets >279 and $<279 \times 10^3/\text{mm}^3$, was 54.73 (95%CI: 47.61–61.85) and 61.73 (95%CI: 55.21–68.26) months, respectively ($p = 0.09$). Likewise, a longer average survival (145.53 months—95%CI: 136.94–154.12) is observed when the PLR < 191.5 (vs. 101.26 months—95%CI: 85.36–117.11 for PLR > 191.5) ($p = 0.01$). Patients with SII < 431.3 had an OS of 130.37 months (95%CI: 117.40–143.33). There was no death for patients with SII > 431.31, and it was not possible to compare.

**Table 4.** Disease-free survival (DFS) and overall survival (OS) of patients with luminal breast cancer, based on the Cox regression analyses.

| | DFS | | | | OS | | | |
| --- | --- | --- | --- | --- | --- | --- | --- | --- |
| | Univariable | | Multivariable | | Univariable | | Multivariable | |
| | *p* | HR Ratio (95% CI) | *p* | HR Ratio (95% CI) | *p* | HR Ratio (95% CI) | *p* | HR Ratio (95% CI) |
| Age | 0.79 | 0.996 (0.64–0.30) | — | — | 0.15 | 0.961 (0.91–4.02) | — | — |
| Grade 3 (vs. Grade 1–2) | 0.01 | 2.754 (1.27–5.98) | 0.53 | 1.346 (0.53–3.42) | 0.23 | 1.992 (0.64–6.18) | — | — |
| T2 (vs. T1) | 0.12 | 2.520 (0.79–8.04) | 0.53 | 1.497 (0.43–5.20) | 0.17 | 4.516 (0.53–38.74) | 0.18 | 4.761 (0.50–45.64) |
| T3 (vs. T1) | 0.01 | 8.118 (2.16–30.48) | 0.13 | 3.221 (0.72–14.40) | 0.02 | 14.337(1.48–148.32) | 0.37 | 2.984 (0.27–32.97) |
| T4 (vs. T1) | 0.00 | 8.622 (2.41–30.80) | 0.19 | 2.713 (0.61–12.18) | 0.03 | 11.686 (1.21–113.03) | 0.54 | 2.206 (0.18–27.16) |
| N1 (vs. N0) | 0.02 | 4.390 (1.32–14.59) | 0.05 | 3.474 (1.00–12.21) | 0.22 | 2.533 (0.57–11.33) | 0.13 | 4.139 (0.65–26.52) |
| N2 (vs. N0) | 0.00 | 18.497 (5.29–64.64) | 0.00 | 11.255 (2.84–44.63) | 0.14 | 3.810 (0.63–22.88) | 0.03 | 21.012 (1.42–311.08) |
| N3 (vs. N0) | 0.00 | 11.838 (3.30–42.42) | 0.01 | 6.396 (1.45–28.18) | 0.02 | 6.561 (1.31–32.81) | 0.01 | 23.234 (2.08–259.71) |
| Platelets ($\times 10^3$/mm$^3$) > 279 (vs. <279) | 0.04 | 2.241 (1.03–4.89) | 0.65 | 1.307 (0.42–4.07) | 0.04 | 4.788 (1.05–21.93) | 0.58 | 1.638 (0.28–9.52) |
| PLR > 191.5 (vs. <191.5) | 0.58 | 1.263 (0.549–2.905) | — | — | 0.01 | 4.402 (1.39–13.94) | 0.00 | 16.156 (2.83–109.25) |
| KI67 Pos (vs. Neg) | 0.16 | 1.728 (0.80–3.73) | — | — | 0.38 | 1.663 (0.53–5.19) | — | — |
| Histology Lobular (vs. Ductal) | 0.53 | 1.410 (0.49–4.09) | — | — | 0.64 | 1.440 (0.31–6.58) | — | — |
| Platelet ($\times 10^3$ mm$^3$) | 0.27 | 1.003 (0.99–1.01) | — | — | 0.22 | 1.004 (1.00–1.01) | — | — |
| Leukocyte | 0.11 | 1.000 (1.00–1.00) | — | — | 0.29 | 1.000 (1.00–1.00) | — | — |
| Neutrophil | 0.25 | 1.000 (1.00–1.00) | — | — | 0.43 | 1.000 (1.00–1.00) | — | — |
| Lymphocyte | 0.14 | 1.000 (1.00–1.00) | — | — | 0.17 | 0.999 (1.00–1.00) | — | — |
| Monocyte | 0.17 | 1.002 (1.00–1.00) | — | — | 0.93 | 1.000 (1.00–1.01) | — | — |

HR: Harzard ratio; CI: confiance interval.

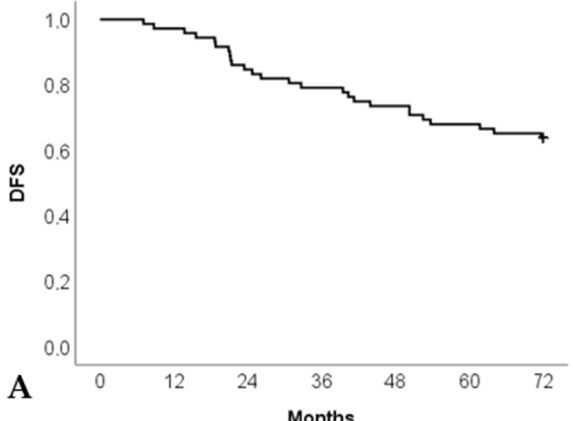

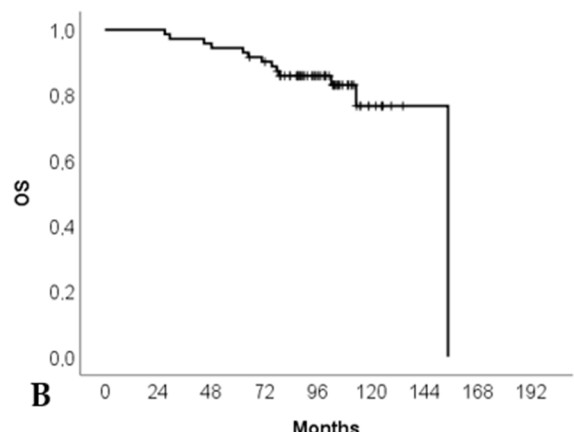

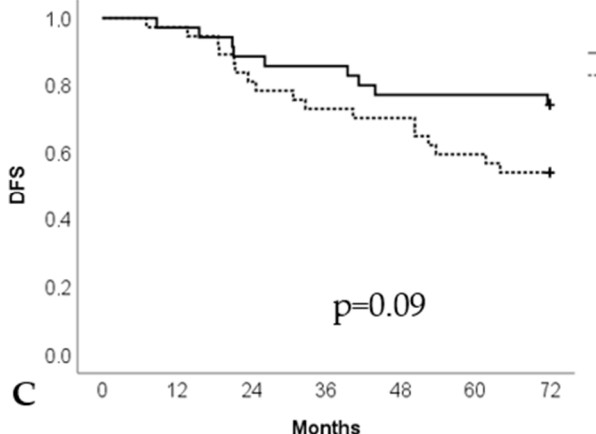

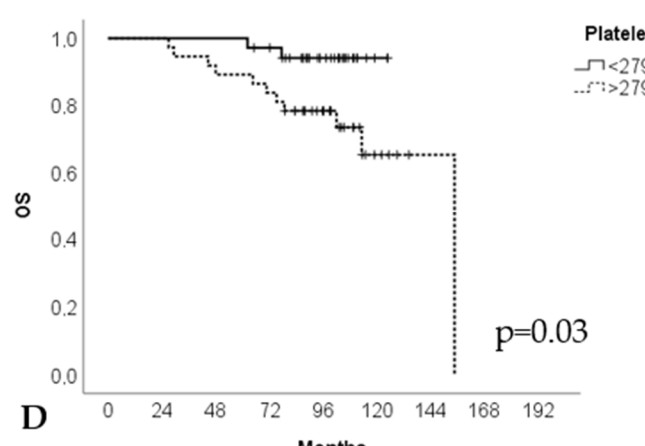

**Figure 1.** *Cont.*

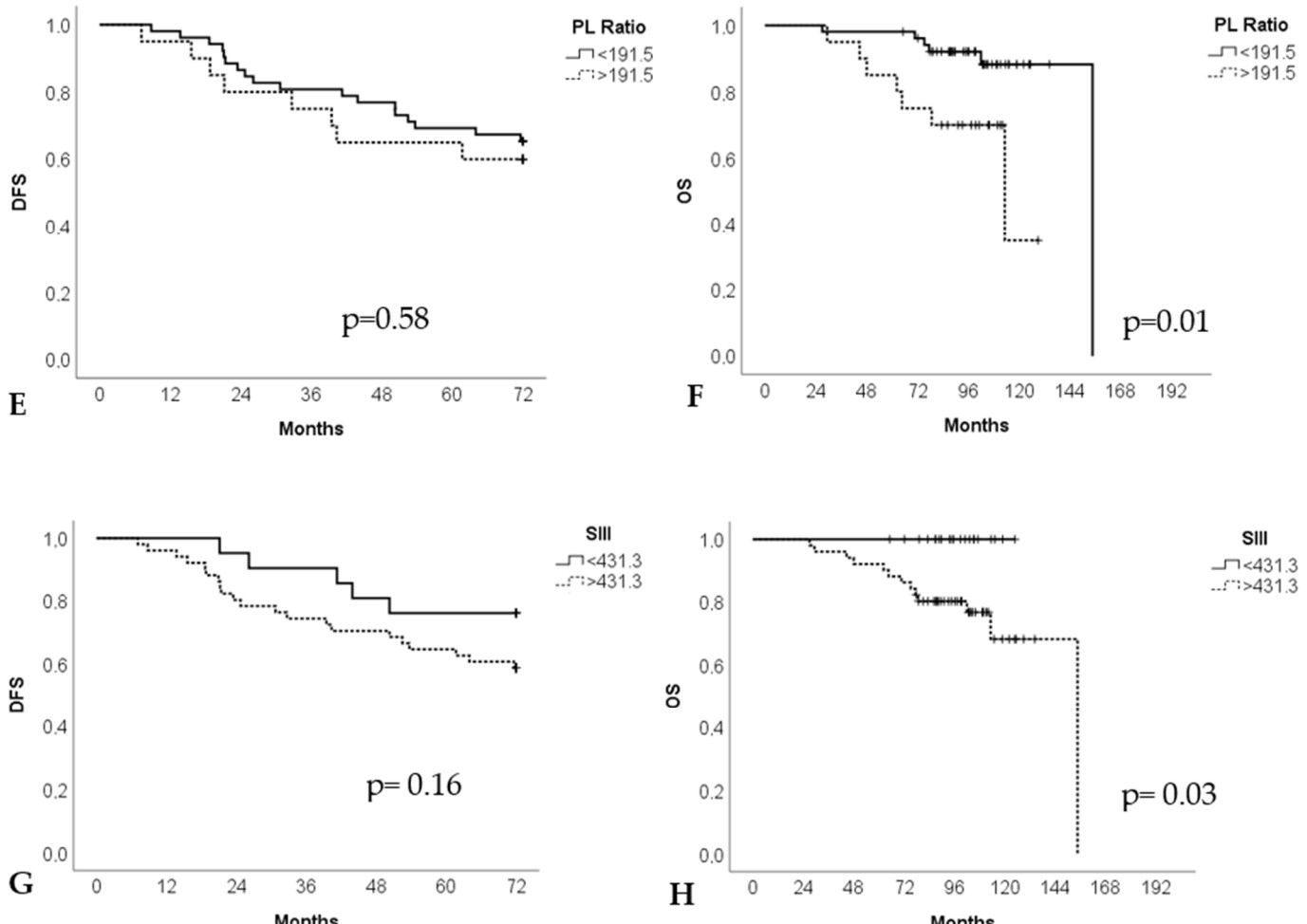

**Figure 1.** Prognostic value of platelet concentration ($\times 10^3/\text{mm}^3$), platelet–lymphocyte ratio (PLR) and systemic index of inflammation (SII) to disease-free survival (DFS) and overall survival (OS) in luminal breast cancer patients. Kaplan–Meier curves to DFS (**A**) and OS (**B**); platelet concentration to DFS (**C**) and OS (**D**); PLR to DFS (**E**) and OS (**F**); SII to DFS (**G**) and OS (**H**).

## 4. Discussion

The evidence suggests an important role for inflammation in the development of cancer, tumor angiogenesis, and metastasis [14]. Chronic oxidative stress and the formation of reactive oxygen species (ROS) caused by the inflammatory response can lead to the initiation and promotion of cancer [5]. As a marker of inflammation, preoperative leukocyte levels are associated with prognosis in several carcinomas. The ratios between different blood cells have been explored between neutrophils/lymphocytes (NLR), monocytes/lymphocytes (MLR), lymphocytes/monocytes (MLR), platelets/lymphocytes (PLR), and the systemic inflammation index (SII), which are routinely available as markers of the systemic inflammatory response and identified as prognostic parameters in several solid tumors [10,15–17]. We performed a cohort study of LBC patients to evaluate the prognostic value of the peripheral neutrophil, lymphocyte, monocyte, and platelet counts, NLR, LMR, PLR, and SII, as well as other clinical factors.

Lymphocytes, macrophages, neutrophils, and platelets play an important role in carcinogenesis, and the tumor controls the immune system in a systemic way. Neutrophils play a significant role in tumor biology, inducing the formation of neutrophils with a pro-tumorigenic phenotype, capable of stimulating tumorigenesis and suppressing the antitumor immune response [8]. The distribution of monocyte chemo-attractive protein (CCL2) in the tumor area can be a critical determinant of monocyte recruitment, as these

are activated in the tumor epithelial region, playing a crucial role in the recruitment of these cells [18]. Tumor-infiltrating lymphocytes (TILs) are important mediators of the antibody-dependent cell-mediated cytotoxicity mechanism and CD8+ and CD4+ T lymphocytes play an essential role in the anti-tumor reaction of the immune system by inducing tumor cells to apoptosis [19].

A direct correlation between circulating tumor cells and inflammatory indices such as RNL, RPL, RML, and SII has been shown, being predictive of OS in metastatic triple-negative breast cancer (TNBC) [10]. Jia et al. (2015) showed that a higher pretreatment peripheral NLR significantly and independently indicated a poor prognosis for BC and TNBC, and this measurement exhibited a greater prognostic value than a lower LMR [20]. The NLR was not a prognostic factor for other BC subtypes. Noh et al. (2013) showed that NLR was an independent prognostic factor for the luminal A breast cancer (BC) subtype; however, NLR can be affected by internal and external factors, including acute and chronic infection and lifestyle-related habits [21,22].

The cutoff values of cellular ratios vary widely due to population characteristics. We defined the cutoff values of our population: platelets show the value of $297 \times 10^3/mm^3$ to the recurrence outcome, and when death was the outcome, platelet values of $279 \times 10^3/mm^3$, 191.5 for PLR, and 431.1 for SII were defined. For LBC, the indices associated with relapse and death outcome were associated with platelet parameters (platelet count and PLR). Patients with platelets $< 297 \times 10^3/mm^3$ had lower clinical staging, no disease recurrence (74.4%), and no death (90.7%). For the death outcome, platelets $< 279 \times 10^3/mm^3$ were associated with lower clinical staging and a lower percentage of patients who died (5.7% vs. 29.7% for >279); 35% death in patients with PLR > 191.5 (vs. 11.5% PLR < 191.5); as well as lower OS (poor prognosis) at 6 years (101.2 vs. 145 months for PLR > 191.5).

Platelets play a fundamental role in cancer progression, as suggested by experimental and epidemiological studies in human cancers including BC [14,23]. Platelet activation is essential for the progressing tumor to promote angiogenesis, extracellular matrix degradation, and the release of adhesion molecules and growth factors [24]. Platelets protect tumor cells from being attacked by the immune system, improving motility and tumor growth. The increase in platelets circulating in the results of the high PLR can be explained by inflammatory cytokines such as IL-3, IL-6, and IL-10 that are released by cancer cells and capable of stimulating the proliferation of megakaryocytes [25]. In our study, patients with higher platelet indices ($>279 \times 10^3/mm^3$ and >191.5) showed a higher mortality rate (29.7% and 35% vs. 5.7% and 11.5%, respectively). Krenn-Pilko et al. (2014) showed a correlation between the absolute value of circulating platelets and the serum level of VEGF (vascular endothelial growth factor), important in tumor angiogenesis [24]. Therefore, an increase in platelet counts reflects an index of tumor-induced inflammation.

LBC shows a high expression level of hormonal receptors (estrogen and progesterone receptors), displaying high TNF-$\alpha$, TGF-$\beta$, and enhanced oxidative stress levels associated with reduced IL-12 levels. BC is an inflammatory disease, and its progression involves changes in redox metabolism, leading to cellular injury and irreversible DNA damage caused by reactive species and cytokines [2,26]. Tamoxifen and its metabolites accumulate in tumors, killing both ER$\alpha$-positive and ER$\alpha$-negative BC cells that depend on oxidative damage, and antioxidants rescue the cancer cells from tamoxifen-induced apoptosis. BC cells respond to tamoxifen-induced oxidation by the activation of the antioxidant response element (ARE) [27].

Enhanced oxidative stress due to estrogen-related mechanisms might cause a condition of persistent platelet activation capable of sustaining BC growth and progression, contributing to tumor invasiveness. In line with the hypothesis of estrogen-induced oxidative stress, patients with TNBC had a reduced oxidant status compared with luminal-like subtypes [1].

In our study, the preoperative values of platelet concentration, RPL, and SII were significantly associated with prognosis in LBC. These statistical associations maintained their importance after adjusting for other possible predictors of patient outcomes such as age

≤40 years, tumor grade, and clinical staging. A PLR > 191.5 was a significant independent predictor of lower OS (HR: 16.156, $p = 0.00$), and a longer average survival (145.53 months) is observed when the PLR < 191.5 (vs. 101.26 months, >191.5) ($p = 0.01$). Azab et al. (2013) studied the impact of PLR on OS in 437 BC patients with quartile categorization and found that patients with the highest PLR quartile (≥185) had a higher mortality rate over 5 years [28]. An increase in RPL pretreatment represented an independent prognostic factor in the time between the diagnosis and the death date related to BC and OS in patients with non-metastatic BC. In luminal subgroup B, PLR was superior in terms of a prognostic impact compared to age and clinical stage T and N [23]. Our data are consistent with the studies reported by Azab et al. (2013) and Krenn-Pilko et al. (2014) that showed that a high RPL, clinical stage, and tumor grade are predictors of BC mortality [24,28].

Studies report that patients with a high density of lymphocytes in the tumor stroma had a better clinical outcome compared to those with a low density of lymphocytes [12,17]. Patients with BC and with high lymphocyte counts had an improvement in survival when compared with those with a lower lymphocyte count [28]. In this study, we did not detect changes in the RNL and RML ratios, but we observe that PLR < 191.5 and a platelet count >279 × $10^3$/mm$^3$ were related to a higher OS, suggesting that the lymphocytes have an important role in the cellular response, suggesting an antitumoral action. Then, a low PLR could be related to an increase in the number of circulating lymphocytes (as suggested by SII and RPL, where lymphocytes are in the denominator and the lower ratio are associated with a higher OS), serving as a factor of better clinical outcome.

LBC has receptors for female hormones, being very responsive to hormonal treatments with tamoxifen. Chemotherapy has little effect on these types of cancers, but it has better effects on luminal B [3,29]. In this study, 18 (29.5%) patients who underwent chemotherapy benefited from the treatment ($p < 0.01$). Von Minckwitz et al. (2013) suggest that the various BC phenotypes require different chemotherapy approaches and show that even patients with LBC can obtain greater benefits from chemotherapy [30].

Nonetheless, the present study is the first study to evaluate the prognostic value of both the platelet count and the PLR in LBC exclusively treated with tamoxifen in the Brazilian population. Our study demonstrated that pretreatment platelet count and PLR are significant predictors of higher OS (platelet > 279 × $10^3$/mm$^3$ and RPL < 191.5) among LBC patients. However, the multivariate analysis did not demonstrate any prognostic significance of platelet concentration for DFS, likely because of the strong predictive ability of PLR affecting the function of the platelet in the multivariate analysis. Based on the multivariate analysis, only the PLR remained an independent significant predictor of OS in LBC patients.

It is suggested that patients with elevated pretreatment RPL may be candidates for approaches that are more aggressive or for more rigorous treatment monitoring, and then the determination of RPL can help to obtain a more accurate individualized risk profile for mortality and contribute to the personalized treatment of BC patients. However, the limitations of this study are associated with either the limited sample number or to the performance of a retrospective analysis that did not allow us to evaluate some outcome and prognosis criteria. The lack of data in medical records caused an important exclusion of patients from the study. It is a single-center study; thus, a multicenter study is required to complement the existing data.

## 5. Conclusions

The preoperative values of RPL and platelet concentration were significantly associated with prognosis in LBC patients. A pretreatment PLR of <191.5 significantly and independently indicated a better prognosis and is an independent predictor for higher OS, while an absolute platelet count >297 × $10^3$/mm$^3$ was associated with higher OS, but not as an independent factor.

**Author Contributions:** Conceptualization, L.N.R. and C.G.B.; methodology, A.D.R.O.R. and M.D.B.; software, A.D.R.O.R., M.D.B. and R.V.A.; validation, A.D.R.O.R. and M.D.B.; formal analysis, A.D.R.O.R., M.D.B. and L.N.R.; investigation, A.D.R.O.R. and M.D.B.; data curation, A.D.R.O.R., M.D.B. and R.V.A.; writing, A.D.R.O.R. and L.N.R.; writing—review and editing, all authors; visualization, A.D.R.O.R. and L.N.R.; supervision, L.N.R.; project administration, L.N.R. and C.G.B.; funding acquisition, C.G.B. and L.N.R. All authors have read and agreed to the published version of the manuscript.

**Funding:** This research was funded in part by the Coordenação de Aperfeiçoamento de Pessoal de Nível Superior—Brasil (CAPES)—Finance Code 001.

**Institutional Review Board Statement:** The study was conducted in accordance with the Declaration of Helsinki and approved by the Institutional Review Board (or Ethics Committee) of Federal University of Health Sciences of Porto Alegre (CAAE 22396713.1.0000.5335) on 10 February 2014.

**Informed Consent Statement:** Patient consent was waived due to it being a retrospective study with the data obtained anonymously.

**Data Availability Statement:** The data that supported the findings in this study are available on reasonable request from the corresponding author. The data are not publicly available due to ethical restrictions.

**Acknowledgments:** We are grateful to Carem Luana Machado Lessa for technical support and to Cristiane Bündchen for statistical support.

**Conflicts of Interest:** The authors declare no conflict of interest.

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
