# Peer review of "Platelet Concentration and Platelet/Lymphocyte Ratio as Prognostic Indicators in Luminal Breast Cancer"

_jmp, doi:10.3390/jmp4010002_

Round 1
Reviewer 1 Report
General Comments:
Several prognosis factors are used to predict the survival of cancer patients, among them are immune cells and platelets. Accumulating evidence supported that the interplay between local immune response and systemic inflammation may play a fundamental role in the development and progression of various cancers. The aim of this study was to investigate the role of platelet concentration and platelet/lymphocyte ratio as prognostic indicators in luminal breast cancer in Brazilian patients with luminal cancer breast cancer treated with tamoxifen.
The manuscript is very well written and organized, data are properly statistically analyzed using Cox proportional hazards models for survival outcomes, and results are clearly presented.
Specific Comments:
1. Abstract: Define LBC (ln 17).
2. Abstract and Introduction: Define the aim and goals of this study.
3. Abstract and text: When use the lower OS, state also a poor prognosis.
4. Mention the specificity of tested population in Abstract too: Brazilian patients with LBC treated with tamoxifen.
5. Material and Methods: State how these values are obtained: manual counts or instruments? Are these techniques and technologies comparable since these data are collected from 2007-2014?
Author Response
Reviewer 1
We thank the reviewer for taking the time to evaluate the submitted manuscript and for the observation pointed to it. Below is the return to the appointments:
- Abstract: Define LBC (ln 17).
We define LBC in line 18, as suggested (presented below)
- Abstract and Introduction: Define the aim and goals of this study.
We adjusted the phrase to clarify the goal of the study in the abstract (lines 16-18), to: “We goal to evaluate the prognostic significance of neutrophil/lymphocyte (NLR), monocyte/lymphocyte (MLR) and platelet/lymphocyte (PLR) ratios in Brazilian patients with Luminal Breast Cancer (LBC) treated with tamoxifen”
- Abstract and text: When use the lower OS, state also a poor prognosis.
Abstract - We opted to present the same information wrote to point the better prognosis: “Both univariate (p=0.01) and multivariate analysis revealed that PLR<191.5 was a significant independent predictor of higher OS/better prognosis (HR=16.16, 95%CI:2.83-109.25, p=0.00)”.
Text - It was added the expression in line 245/246 “35% death in patients with PLR>191.5 (vs 11.5% PLR<191.5), as well as lower OS (poor prognosis) in 6 years (101.2 vs 145 months for PLR>191.5).
- Mention the specificity of tested population in Abstract too: Brazilian patients with LBC treated with tamoxifen.
We adjusted according to the reviewer's suggestion (as presented in item 2).
- Material and Methods: State how these values are obtained: manual counts or instruments? Are these techniques and technologies comparable since these data are collected from 2007-2014?
We acknowledge the reviewer by the observation, and we added the information in lines 76-78: “…was used and the absolute values ​​of neutrophils, lymphocytes, platelets, and monocytes were collected (obtained by automatized hematological technologies comparable, during all the period of the study)”.
English revision:
Some mistakes were corrected in the text. Some examples:
Line 137 – “No death was observed in patients who presented SII<431.3, while 25.5% of patients died when SII>431.3.”
Table 2 (footnote): The results are expressed as n (%), except for the absolute count of leukocytes, neutrophils, lymphocytes, monocytes, and platelet (x103/mm3), which are expressed as media±standard deviation. We also revised the tables, and some commas were substituted by points, observing the English language.
OBS: To attend to the suggestions/observations pointed by another reviewer, some other points were altered and are marked in the text.

Reviewer 2 Report
While the topic and some of the findings are interesting, this paper does not have the quality required for publication in the current version. It has results of data analysis, but the presentation and narrative of data and findings lack good logical order and sometimes are confusing. A revision with data that is clearly and directly supportive of the conclusions would be helpful.
Author Response
Reviewer 2
We thank the reviewer for taking the time to evaluate the submitted manuscript. We revised the manuscript to identify points that could be improved, according to the evaluation of the criteria established by the journal and pointed out by the reviewer. In the text are signalized the modifications and the comments are presented below. We are receptive to suggestions to improve the manuscript.
a) Introduction
We revised the introduction observing a didactic construction and the information that is presented, as well as the references that support it. Reference number 9 was substituted by a more actual study, and that adds the use of leukocyte parameters as a prognosis marker to metastasis in Luminal Type Breast Cancer.
b) Research design
The presentation of the results followed the established: First, are presented the characteristics of the LBC patients according to the outcomes (relapse or death) in Table 1. After, we made the ROC curves for the studied markers and later we observed the Kaplan Meier curves to outcomes DFS and OS, to the index’s platelet concentration, PLR and SII, that showed AUC >0.6 in ROC curves. The baseline characteristics of the patients according to the platelet count (Table 2) and according to PLR, platelet count, and systemic index of inflammation (SII) (Table 3), considering the outcome of relapse and death, respectively. Next, the prognostic analyzes are presented (univariate and multivariate analyzes) using Cox's regression model (Table 4).
In the description of the results, those that had a statistically significant difference and that are relevant, according to the objectives of the study to the established outcomes, are highlighted. We revised the text searching for points to be improved and we transfer the information “We did not find difference in OS 6 years between patients with platelets < or >297 (119.1 and 127.4 months, respectively) (p=0.08) (Table 4)” from lines 131/132 (in the prior version of the manuscript) to lines 147-148 (new version)
If possible, I ask for indications of points to be adjusted in the research design, so that the manuscript can be improved. We are open to suggestions. At this moment, we did not significantly change, since the line followed for the presentation of the results is judged comprehensive by us and was considered adequate by the other reviewer “The manuscript is very well written and organized, data are properly statistically analyzed using Cox proportional hazards models for survival outcomes, and results are clearly presented”.
c) Methods
We carefully review this section and added information in lines 74-79: “For the performance of cellular blood analysis, preoperative blood sample closest to the date of surgery (minimum:1 day and maximum: 90 days) was used and the absolute values ​​of neutrophils, lymphocytes, platelets, and monocytes were collected (obtained by automatized hematological technologies comparable during all the period of the study), which were used to establish the ratio between cells and the systemic inflammation index (SII).
d) Conclusion
We acknowledge the reviewer by the observation. We revised the conclusion and are in accordance with his/her observation. Because of this, the conclusion was adjusted based exclusively on the results of this study.
Previous conclusion: A higher pretreatment PLR significantly and independently indicated a poor prognosis and is an independent predictor to the higher OS, while absolute platelet count (>297x103/mm3) was associated to higher OS, but not as an independent factor. The index can help in support to oncological therapy decisions, suggesting that cancer-associated platelet ratio may play a greater role in promoting breast cancer progression in the luminal subtype.
Substituted by: Preoperative values of RPL, and platelet concentration were significantly associated with prognosis in LBC. A pretreatment PLR (<191.5) significantly and independently indicated a better prognosis and is an independent predictor for the higher OS, while absolute platelet count (>297x103/mm3) was associated to higher OS, but not as an independent factor.
- e) English revision
Some mistakes were corrected in the text. Some examples:
Line 137 – “No death was observed in patients who presented SII<431.3, while 25.5% of patients died when SII>431.3.”
Table 2 (footnote): The results are expressed as n (%), except for absolute count of leukocytes, neutrophils, lymphocytes, monocytes, and platelet (x103/mm3), which are expressed as media±standard deviation. We also revised the tables, and some commas were substituted by points, observing the English language.
OBS: To attend to the suggestions/observations pointed out by another reviewer, some other points were altered and are marked in the text.
Please see the attachment.

Round 2
Reviewer 2 Report
Thanks for the revision